# Immunosuppressant Therapies in COVID-19: Is the TNF Axis an Alternative?

**DOI:** 10.3390/ph15050616

**Published:** 2022-05-17

**Authors:** Yadira Palacios, Leslie Chavez-Galan

**Affiliations:** Laboratory of Integrative Immunology, Instituto Nacional de Enfermedades Respiratorias “Ismael Cosío Villegas”, Mexico City 14080, Mexico; yadpal@gmail.com

**Keywords:** TNF, TNFR1, COVID-19, inhibitor, therapy

## Abstract

The study of cytokine storm in COVID-19 has been having different edges in accordance with the knowledge of the disease. Various cytokines have been the focus, especially to define specific treatments; however, there are no conclusive results that fully support any of the options proposed for emergency treatment. One of the cytokines that requires a more exhaustive review is the tumor necrosis factor (TNF) and its receptors (TNFRs) as increased values of soluble formats for both TNFR1 and TNFR2 have been identified. TNF is a versatile cytokine with different impacts at the cellular level depending on the action form (transmembrane or soluble) and the receptor to which it is associated. In that sense, the triggered mechanisms can be diversified. Furthermore, there is the possibility of the joint action provided by synergism between one or more cytokines with TNF, where the detonation of combined cellular processes has been suggested. This review aims to discuss some roles of TNF and its receptors in the pro-inflammatory stage of COVID-19, understand its ways of action, and let to reposition this cytokine or some of its receptors as therapeutic targets.

## 1. Introduction

COVID-19 is an infectious disease caused by the Severe Acute Respiratory Syndrome coronavirus-2 (SARS-CoV-2); it was declared a pandemic by the World Health Organization (WHO) in March 2020 [1,2,3]. On 18 March 2022, the WHO reported 462,58,117 confirmed cases worldwide and 6,056,725 deaths attributed to COVID-19 [4].

Epidemiological evolution shows that of those SARS-CoV-2 infected, about 80% present mild or moderate disease, 15% of patients developed a severe disease requiring oxygen, and near to 5% evolve to critical disease where complications such as respiratory failure, acute respiratory distress syndrome, sepsis, and septic shock, thromboembolism, and even multi-organ failure can manifest [5]. To date, five VOC (variants of concern) to SARS-CoV-2 have been described: Alpha (B.1.1.7), Beta (B.1.351), Gamma (P.1), Delta (B.1.617.2), and Omicron (B.1.1.529), reports indicate that the Omicron infection induces the lower risk of severe disease, but it has been considered highly transmissible [6,7]. Currently, millions of people receive vaccine doses worldwide (around 10.869.655.945 doses have been administered on 18 March 2022) [4], but unfortunately, we do not have an optimal therapy for COVID-19, mainly to use in moderate to critically ill patients. Therefore, scientists worldwide join forces to understand the clinical significance of molecular events that may lead some patients to display mild symptoms or even asymptomatic while others develop a severe clinical course that may culminate in death. Although diverse therapeutic proposals have been considered, the reports of adverse events attributable to the therapy are increasing [8]. Thus, the current emergency to control the evolution from moderate-severe COVID-19 to death should propose the use of therapeutic schemes beyond what concerns antivirals only.

Since the beginning of the pandemic, cytokine levels emerged as indicative of disease progression, suggesting that the pro-inflammatory cytokines play a determining role in the pathogenesis of COVID-19. The pro-inflammatory cytokines modify the immune system and, if it is out of control, in some cases, lead to fatal consequences [9]. The COVID-19 severity is associated mainly with the development of a phenomenon called "cytokine storm," characterized by an intense inflammatory process that fails to be regulated; this phenomenon could happen both under infectious or non-infectious contexts [9,10,11].

The absence of regulatory mechanisms to modulate the inflammatory process is responsible, at least partially, for favoring a cytokine storm that progresses to organ dysfunction and even death [9,12,13]. The multicomponent profile called "storm," typical of the innate viral immunity, includes cytokines, chemokines, and reactive oxygen species [14]. Tumor necrosis factor (TNF) is one of the most critical pro-inflammatory cytokines produced by several cells such as monocytes/macrophages, natural killer (NK), neutrophils, and T-cells, among others; it is expressed mainly in the acute phase of infections and is considered a key regulator of the immune response [15]. TNF triggers complex signaling where a low level benefits the health, but high levels can promote organ damage [15,16,17]. As in all cases of ligand-receptor interactions, the function of this cytokine is closely linked to its receptors. Thus, although the effect of one cytokine does not trigger COVID-19 complications, the TNF axis demands significant attention [18,19].

## 2. More than One Rout Favor the Cytokine Storm in COVID-19

The concept “cytokine storm” was published for the first time in 1993 in graft-versus-host processes; later, it was appropriated in viral and some bacterial infections [11]. It has been described in diverse infectious and non-infectious diseases, but in 2020 it gained attention in the context of the COVID-19 pandemic, alluding to an exaggerated release of pro-inflammatory cytokines that leads to tissue damage [14].

Under the context of non-infectious diseases, such as those with an autoimmune origin that display a robust inflammatory process such as systemic lupus erythematosus and systemic juvenile idiopathic arthritis, the cytokine storm is associated with macrophage activation syndrome (MAS) [20]. In this context, some of the cytokines that play an essential role as a member of the cytokine storm are interleukin (IL)-6, interferon-gamma (IFN-γ), TNF, IL-1β, and IL-18, as well as some chemokines [21].

In infectious diseases such as COVID-19, the cell entry used by the virus impacts the activation of the immune response; this is related in particular to the receptors used [13]. The angiotensin-converting enzyme 2 (ACE2) is the dominant SARS-CoV-2 receptor; although coronavirus infection influences the ACE2 expression, other host-dependent conditions may impact downregulation and malfunction, such as single nucleotide polymorphisms (SNPs) and a phenomenon of compartmentalization has also been suggested [22,23,24,25]. Once the SARS-CoV-2 arrives at the pulmonary alveoli, the head of the S protein is targeted by host proteases as the type II transmembrane serine protease (TMPRSS2), which generates S1 and S2 subunits; this activation leads the S1 subunit to interact with ACE2 [26]. Then, a conformational change in the S2 subunit leads to viral/host membranes fusion to enter the cell. Other proteases such as Neuropilin1 (NRP1), cathepsin L, cathepsin B, trypsin, factor X, elastase, and furin have been described as co-factors that facilitate this process. Here is important to mention that the alveolar epithelial type 2 cells express a higher level of ACE2 in the lung; consequently, they are predominantly infected [14,27,28,29,30].

ACE2 is shedding on the membrane cell surface by the effect of TMPRSS2 and the disintegrin and metalloprotease 17 (ADAM17), and this last also is responsible for TNF, TNF receptor 1 (TNFR1), and TNF receptor 2 (TNFR2) shedding [31,32]. Reports suggest that ADAM17 and TMPRSS2 compete for ACE2, although they are cleaved at different sites, and apparently, when ACE2 is cleaved by TMPRSS2, it favors the SARS-CoV-2 cell entry, while ADAM17-cleaved ACE2 has the opposite effect because seemingly ACE2 produces soluble ACE2 that may neutralize SARS-CoV-2 infection. However, ACE2-shedding by TMPRSS2 may block the ADAM17-cleavage activity [32,33,34,35]. It is well documented that for both SARS-CoV and SARS-CoV-2 infection, TMPRSS2 efficiently promotes virus-plasma membrane fusion [34], a mechanism more efficient for viral replication than endocytosis mediated by ADAM17-cleaved ACE2 [32].

Once the virus is inside the cell, the activation of cell death mechanisms such as apoptosis, necroptosis, pyroptosis, and PANoptosis induce the release of pathogen-associated molecular patterns (PAMPs) and danger-associated molecular patterns (DAMPs), which are recognized by pattern recognition receptors (PRRs), mainly Toll-like receptors (TLR); molecules expressed on natural killer cells, dendritic cells, macrophages, and neutrophils [12,14,36,37] (Figure 1A).

In the infection site, chemoattractants, such as CCL2 and CXCL8, are produced to help in cell mobilization as monocytes and neutrophils, which can also induce inflammatory mediators; epithelial and endothelial cells also contribute to both inflammation and induction of cell death mechanisms [37,38]. Single-cell immune analysis in bronchoalveolar lavage (BAL) suggested that mild COVID-19 has mainly anti-inflammatory macrophages; in contrast, in critical illness, hyperinflammatory macrophages promote the release of cytokines and chemokines like IL-6, IL-8, IL-1β, TNF, CXCL10, CCL8, CCL20, CXCL2, CXCL3, CCL3, CXCL3, and CCL4 [38,39] (Figure 1B).

Another consequence that maintains a constant inflammation status in the infection site is the inflammasome activation by the generated PAMPs and DAMPs. NLRP3 is the central protein platform for caspase-1 activation, leading to the proteolytic maturation of IL-1β and IL-18. The inflammasome induces an inflammatory cell death termed pyroptosis [12,36,40,41] (Figure 1C). IL-18 also acts as neutrophil-attracting to favor its continuous infiltration in infected lungs; thus, neutrophils are activated by IL-17, and they can form neutrophil extracellular trap (NET), a cell death type precisely in neutrophils, and it is called NETosis, where several intracellular components are released as neutrophil elastase, cathepsins, lactoferrin, myeloperoxidase, and granule proteins. Moreover, mitochondrial DNA, proteins derived from the cytoplasm and cytoskeletal, and DNA with histones may be present. NETosis has been reported as a relevant event to the immunopathology of thrombosis, apoptosis, and organ damage [42,43] (Figure 1D). The pro-inflammatory behavior produced by the events described helps T-cell recruitment, and these cells produce TNF and IFN-γ, impacting endothelial and dendritic cells [43] (Figure 1E).

Although the initial reports indicated a relationship between the severity of the disease and high inflammatory cytokine levels, recent evidence questions whether COVID-19 activates a cytokine storm. Some authors described that pro-inflammatory cytokines levels are not higher than in other pathologies with acute respiratory distress syndrome (ARDS), and others suggest that there are lower levels in COVID-19 than those observed in non-COVID-19 pathologies [10,44,45,46]. Mainly, TNF, IL-6, and IL-8 were lower in COVID-19 compared to patients with septic shock/ARDS, and due to IL-6 and IL-8 was near eightfold lower, it was suggested that cytokine storm does not characterize COVID-19, and it could be one of the reasons for the immunosuppressive therapies fail [45]. Even when there is a discrepancy in deciding whether patients with COVID-19 activate a true cytokine storm, the severity of the disease is characterized by an inflammatory process that needs to be modulated or stopped to have a better prognosis in patients.

## 3. Relevant Targets of the Inflammatory Response for Immunotherapy in COVID-19

Several efforts have focused on identifying cytokine profiles that predict a poor outcome to avoid mortality. In this regard, at the beginning of the pandemic, several reports converged in identifying increased levels of individual cytokines like IL-6 as potential biomarkers or targets for therapy. However, it later was proposed that the impact of the combined effect of two or more cytokines such as IL-6, IL-1β, TNF, IFN-γ, and CXCL10, among others, must be considered [9,12]. IL-6 has been one of the first references and therapeutic candidates [47]. Furthermore, various clinical protocols are carried out to evaluate the efficacy of anti-cytokine inhibitors as potential therapeutic options [1,48,49,50,51,52]. Nevertheless, the mechanism of action is not fully understood, in addition to the contradictory or adverse events observed [8].

The conventional approved immunotherapy to mediate neutralizing the inflammatory function through monoclonal antibodies (mAbs) has been used in different pathologies [53]. mAbs are produced by one B-cell clone and represent an efficient therapeutic intervention against several diseases because they offer strict specificity and a strong affinity to the target. As a kind of passive immunotherapy, mAbs act through different mechanisms, being the principal blockers of the target and neutralizing its function [54]. In the COVID-19 context, one of the first targets was IL-6 because reports showed that high serum levels of this cytokine correlated with severity [55]. IL-6 interacts with the IL-6 receptor (IL-6R) in the transmembrane or soluble form; after the ligand-receptor interaction, IL-6R complexes with gp130 membrane-bound to facilitate the signaling [55,56,57]. In addition, the IL-6/IL-6R interaction activates the Janus kinase (JAK) signaling downstream to mediate diverse functions in the immune cells. Thus, IL-6R represents an excellent therapeutic target; IL-6R-blocking mAbs such as tocilizumab and sarilumab have been included in the NIH COVID-19 Treatment Guidelines indicated in hospitalized adults that require oxygen delivery through a high-flow device or noninvasive ventilation [58]. IL-6R blockade may represent an essential strategy for interrupting the inflammatory process in COVID-19; however, the success has been limited [56]. In this regard, probably some of the reasons why this has been limited are because it has been reported that COVID-19 patients display variable levels of IL-6, and the association of secondary infections with immunomodulators such as tocilizumab also is a common risk that has been identified [47,56,57,59]. In summary, at present, we do not know exactly how is the mechanism of the inflammatory response against the SARS-CoV-2, and this is essential knowledge to develop therapies to suppress the uncontrolled inflammatory process without side effects.

Worldwide, there are around eleven antibodies targeting SARS-CoV-2 or modulators of the deregulated immune response that have received approvals both as emergency use or authorized [60]. The combined administration of anti-SARS-CoV-2 spike protein mAbs, bamlanivimab, and etesevimab, was indicated in mild-COVID-19 patients with a high risk of evolving into severe disease [60,61]. Sotrovimab is a human neutralizing anti-SARS-CoV-2 antibody approved in Australia for hospitalized patients with an increased risk of death [62,63]. Another antibody is regdanvimab (regkirona), which targets the SARS-CoV-2 viral spike protein and was approved in South Korea to treat mild-to-moderate COVID-19 in patients older than 50 with chronically diseases or immunosuppressed [60,64]. Two more combined mAbs with emergency use authorization in USA and Australia for high-risk outpatients non-hospitalized are casirivimab plus imdevimab for treatment and COVID-19 prevention [60,65].

Anakinra is another mAb used to modulate the pro-inflammatory response; it is an interleukin receptor antagonist that binds to the IL-1 type I receptor; it has been considered in combination with tocilizumab for COVID-19 clinical trials [66,67]. In the same way, canakinumab, a neutralizing mAb of IL-1β through competition for IL-1RI binding, is used to treat auto-inflammatory severe diseases and has been evaluated in COVID-19 patients [68,69,70,71]. Focusing on the signaling of cytokines implicated in COVID-19, the Janus-associated kinase (JAK)-inhibitor, baricitinib, has been suggested in combination with remdesivir in hospitalized COVID-19 patients [72,73]. Baricitinib is a synthetic inhibitor of the isoforms JAK1/JAK2; this drug shows few drug–drug interactions and is currently used in rheumatoid arthritis when the anti-TNF therapy fails [74]. Baricitinib affects the production of cytokines like IL-2, IL-6, IL-10, IFN-γ, and granulocyte-macrophage colony-stimulating factors during the inflammatory process and has been suggested that it may exert anti-viral effects inhibiting the endocytosis of SARS-CoV-2 [72,74]. The use of remdesivir in combination with baricitinib, on one side, prevents the hyperinflammatory response, and on the other, there is an advantage of the broad-spectrum antiviral that remdesivir presents. A double-blind study reported that COVID-19 hospitalized adults had higher safety than those who used remdesivir alone [73]. Interestingly, the incidence of adverse events such as thromboembolic was lower, and patients receiving high-flow oxygen or noninvasive mechanical ventilation improved faster with the combined therapy [72,73].

Thus far, therapeutic antibodies targeting the virus, some pro-inflammatory cytokines, or one of their receptors are highlighted as a focus of attention in COVID-19.

## 4. What Is TNF and Its Inhibitors?

TNF is one of the most critical pro-inflammatory cytokines of the innate immune response and mediates pleiotropic effects, which implies action on diverse cells subpopulations to mediate a wide range of activities such as the production of inflammatory mediators, cell proliferation, and cell death. TNF is produced by macrophages, T-, B- NK-, dendritic cells, and fibroblasts [75]. TNF is a versatile cytokine that acts as an alarm system in host defense, appearing in the first few minutes of damage [76]. TNF is a trimeric molecule that can be found as a transmembrane (tmTNF) or soluble form (sTNF) by the action of ADAM17 [77,78]. Both forms are bioactive molecules after interacting with one of its receptors, TNFR1 or TNFR2 [77,79]. Importantly, the soluble formats of both receptors can also be generated by the sheddase activity of ADAM17. Transmembrane TNFR1 and TNFR2 signaling to active nuclear factor-kappa B (NF-κB) or MAP kinase family inducing cell survival or cell death [80]. When tmTNF interacts with soluble TNFRs may trigger an activation called reverse signaling, which is activated in the tmTNF expressing subpopulation; in particular, sTNFR1 induces apoptosis through tmTNF by reverse signaling [78].

In addition to the ADAM17 activity, the soluble formats of TNFR1 and TNFR2 may result from alternative splicing [81]. The soluble receptor variants have been associated with different biological effects. For example, in viral and bacterial infections, some pathogens can encode soluble homologs of receptors as a mechanism of immune evasion throughout the sequestration of the corresponding cytokine [82,83,84].

Anti-TNF therapy was approved by the U. S. Food & Drug Administration (FDA) in 1998, and currently, it is efficiently administered as a treatment for diverse inflammatory processes like rheumatoid arthritis, juvenile idiopathic arthritis, ankylosing spondylitis, Crohn´s disease, plaque psoriasis, among others [15,76,85,86].

There are five types of biological TNF inhibitors approved for use; the difference between them is the nature of their origin (Figure 2). The first of these TNF inhibitors is infliximab, approved in August 1998, which is an anti-human mAb chimeric mouse-human IgG that inhibits soluble TNF in both the monomeric and trimeric form [77,87]. The second TNF inhibitor, etanercept, approved in November 1998, is a fully human recombinant molecule consisting of two subunits of the TNFR2 linked to a human IgG1 Fc and can block the trimeric form of TNF [77,88]. Third, Adalimumab and Golimumab are fully humanized IgG1, the first was FDA-approved in 2002, the second in 2009, and both neutralize tmTNF and sTNF [77,89]. Finally, Certolizumab-pegol is a PEGylated Fab fragment, which means an IgG1 mAb with a hinges region linked to two cross-linked chains of a 20-kDa of polyethylene glycol (PEG) but has lacked the Fc region. It was approved in 2008 and also can neutralize both tmTNF and sTNF [85,89,90].

## 5. Could TNF Be Considered a Target in COVID-19?

The anti-TNF biological agents target both tmTNF and sTNF, and all of them trigger diverse effects such as neutralization, apoptosis, and modulation of the immune system; several facts may affect the efficacy of every anti-TNF inhibitor as pharmacokinetic, tissue penetration, affinity, and avidity, among others [91], reasons by which more systematic reviews are required to understand the global impact of anti-TNF therapy in COVID-19. Despite the benefit of anti-TNF therapy in a broad panel of inflammatory diseases, it has been well documented that the TNF inhibitors may present controversial effects in non-responder patients; this means patients who previously responded to the treatment but posteriorly, they are treatment-refractory and susceptible to infection from some bacteria such as *Legionella pneumonia* and *Listeria monocytogenes* [15,85]. Even more, the side effects in some cases may also include *Mycobacterium tuberculosis* and hepatitis B reactivation, anti-TNF inhibitor antibodies, and neurological impact [15,92,93]. In some cases, the side effects may be overcome by stopping the administration or changing the inhibitor agent.

Various pronouncements have been made to support the administration of anti-TNF agents in COVID-19 patients; they are based on the principle that the increase of this cytokine has severe effects on diverse cell subpopulations and its function blocked with the anti-TNF therapy is efficient in diverse autoimmune diseases [18,19,48,94]. However, the first problem found is the discrepancy in the TNF levels identified in COVID-19 by several groups; some authors showed that COVID-19 patients display increased levels of this cytokine, but others cannot find it [44,95,96,97].

At present, there are ongoing clinical protocols to identify the efficiency of the anti-TNF therapy in COVID-19, some of them suggesting that COVID-19 patients treated with anti-TNF agents have a better prognosis [98,99]. In this regard, recently, it was suggested that patients with any inflammatory disease using TNF inhibitors had a lower probability of hospitalization or the development of severe COVID-19 compared to patients diagnosed with an inflammatory disease but another treatment [98,100]. However, despite the evidence, the use of these drugs is debatable in some cases, and probably it is favored because the knowledge about levels and regulation mechanisms of TNF, TNFR1, and TNFR2 in COVID-19 is limited, and other important factors call into question what side effects can leave the use of TNF immune modulators in viral infections [99].

In 2020, the Crohn´s & Colitis Foundation of America recommended that patients with Inflammatory Bowel Disease (IBD) that develop COVID-19 stop the anti-TNF administration (biologics or biosimilars) until they recover. It was supported by the National Scientific Advisory Committee of the USA, the International Organization for the Study of Inflammatory Bowel Diseases (IOIBD), and the American Gastroenterological Association [101,102,103,104]. These recommendations were made because patients with chronic inflammation receive immunosuppressive or immune-modulator therapies that may represent a risk of viral infections. Additional to the lack of knowledge about what side effects can leave the use of TNF immune modulators in viral infections [103,104].

Recently, Keewan et al. suggested that PEG-, non-PEG-Certolizumab, and Adalimumab favor the infection of SARS-CoV-2 because the mode of action of these agents induces the Notch-1 signaling pathway promoting a pro-inflammatory environment [105]. In addition, previous reports documented that ADAM17 participates in the Notch signaling pathway, which also may lead to an inflammatory response [106,107]. In an in-vitro model of *Mycobacterium avium* subspecies paratuberculosis (MAP) infection has suggested that the Notch-1 signaling pathway induced by the anti-TNF therapy impacts IL-6 and MCL-1 expression in macrophages. Additionally, the authors showed that, in particular, adalimumab increases the TMPRSS2/ADAM17 ratio, suggesting the infection facilitated by TMPRSS2 [105].

Although successful cases of immunotherapy with TNF inhibitors have been well documented for years, indeed, some adverse events are also known. A meta-analysis documented that the use of adalimumab, golimumab, infliximab, certolizumab, and etanercept, significantly increases fungal, viral, and bacterial (including mycobacterial) infections [108]. Demyelinating disorders such as type I diabetes and psoriasis also have been reported as a side effect associated with the use of TNF inhibitors; it is the primary failure that occurs in one-third of patients [109].

Epigenetic control of the *TNF* gene might direct or predict the patient’s responsiveness to anti-TNF therapies, and TNF signaling pathways may be restricted to epigenetic regulation [110]. In particular, the presence of increased levels of H3K4me2 (histone H3 lysine 4 dimethylation) is a modification identified in promoters of expressed genes of TNF-producing cells [111]. The *TNF*-308G>A polymorphism in the promoter region could be associated with a poor response to TNF inhibitors [112]. At the level of the TNF locus, loss of CpG methylation has been reported as an age-associated factor in healthy donors [113]. In this region, a −238 position polymorphism disturbs a CpG motif, a fact that is also related to the severity in cases of arthritis rheumatoid [114].

In the case of COVID-19, there are two main unknowns related to TNF inhibitors: the risk of immunosuppression in the SARS-CoV-2 infection and the possibility that these agents could facilitate virus entry into target cells. Therefore, further molecular and clinical studies are necessary to understand the real potential of anti-TNF inhibitors in COVID-19. Meanwhile, novel treatment targets must be considered for evaluation, alone or combined with antiretrovirals [115]. In this regard, TNFRs deserve particular attention in COVID-19.

## 6. Could Be sTNFR1 a Treatment Target for COVID-19?

Scientific evidence shows that the TNF is not the alone-worker in COVID-19; data indicate that the couple TNF/IFN-γ synergizes to induce crosstalk of several types of cell death such as pyroptosis, necroptosis, and apoptosis (collectively named PANoptosis). It is regulated by signal transducer and activator of transcription 1 (STAT1), interferon regulatory factor 1 (IRF1), inducible nitric oxide synthase (iNOS), and nitric oxide (NO); and Caspase-8/RIPK3, all together induce an inflammatory cell death that is in detriment to the host. Under the COVID-19 context, it has also been identified that this TNF/IFN-γ synergism could be blocked with the mAbs co-administering [12].

At present, it is well known that TNFR1 signaling is related to apoptotic cell death, but additionally, this receptor also promotes the release of pro-inflammatory molecules, including cytokines, chemokines, adhesion molecules, and matrix metalloproteinases [116,117]. Thus, TNFR1 signaling is highly inflammatory, a high level of sTNFR1 in fluids has been previously proposed as a biomarker in several diseases with a pro-inflammatory profile as diabetic nephropathy, hyper-inflammation on the ocular surface, sepsis, ventricular dysfunction, and myocardial infarction, acute lung injury, smokers with microvascular complications, and acute graft-versus-host disease [116,118,119,120,121,122].

Efforts have been made to identify if sTNFR1 could represent a potential biomarker in COVID-19. A first report indicated that COVID-19 patients in an intensive care unit (ICU) have a higher level of a pro-inflammatory profile (IL-1β, IL-6, IL-8), including high sTNFR1 levels [123]. Posteriorly, Mortaz et al. [124] identified high levels of sTNFR1 in patients with severe COVID-19, suggesting for the first time that this molecule could represent a biomarker of severity and mortality. In accordance, other authors identified that sTNFR1 is increased in severe COVID-19 compared with mild and moderate illness, and also it was associated with mortality [97]. However, Bowman et al. [125] reported that both TNFR1 and TNFR2 increased independently of severity.

The use of anti-TNF therapy requires an integrative comprehension considering the roles of TNFR1 and TNFR2, as well as the soluble or transmembrane formats of the proteins [79]. Based on the discussed evidence, we suggest that the clinical trials to evaluate the efficiency of the anti-TNF therapy should do an integral association between levels of TNF and its receptors. Moreover, a relevant question that has not been clarified is whether COVID-19 TNF is found mainly in transmembrane or soluble forms. In other pathologies, it has been deeply described that the TNF form is fundamental to determining its function as an activator or regulator of the inflammatory response [79,126].

Although the current TNF inhibitors are highly efficient, it has been well documented that total TNF inhibition may lead to severe side effects such as tuberculosis reactivation [127,128]. An advantage is that TNFR1 and TNFR2 mediate different cellular events, the reason why the option of selectively blocking one of them may result in fewer adverse effects [129,130]. As TNFR1 promotes inflammation and cell death, the specific inhibition can avoid side effects associated with TNF inhibitors and therefore constitute the next generation of immunomodulators even in viral infection as COVID-19 [131]. Various efforts are committed to the selective modulation of TNF receptors, with TNFR1 receiving greater attention due to the roles in which it has been identified [132,133]. This positions TNFR1 as a potential therapeutic target during SARS-CoV-2 infection. In Table 1, we summarized the human TNFR1 inhibitors that are currently under evaluation; this information is fundamental for developing effective therapeutic strategies for COVID-19.

## 7. What We Know from Preliminary Results about TNFR1-Selective Inhibitors under Progress?

TNFR1 inhibitors include a wide range of molecules, including antibodies, small molecules, and aptamers, that selectively inhibit TNFR1 but without affecting TNFR2 function. The first group of inhibitors includes derivatives of antagonist mAb that interact with the cysteine-rich domain 1 of TNFR1, competing with high affinity with the natural ligand [135,136]. To improve the ability of the Atrosab to binding to TNFR1, the mAb’ humanized variable domains were re-engineering, and the Fv13.7-Fc molecule displayed improved pharmacokinetic properties and antagonistic activity, but at present, this molecule has been not clinically evaluated [136]. Moreover, using inflammatory murine models, atrosimab was highly effective at suppressing the development of chronic diseases like arthritis, non-alcoholic steatohepatitis, and experimental autoimmune encephalomyelitis [151].

Muteins are mutant proteins of TNFR1-selective antagonists; structurally, they have peptide-linkers to enhance their stability and bioactivity and polyethylene glycol (PEG) modification to limit the reactive sites. These mutants showed good avidity and thermal stability for selective binding to TNFR1; using the arthritis murine model, muteins in plasma have an extended half-life, and regulatory T cells are increased compared to etanercept [138,139,140,141,142,152].

Though High-Throughput Screening (HTS) has been developed to identify selective small molecules TNFR1 [145,146]. Lo Ch et al. (2017) found non-competitive inhibitors, such as zafirlukast and triclabendazole, these molecules stabilize the nonfunctional conformational state of TNFR1, impacting IκBα degradation and NF-κB activation [145].

Another example of TNFR1 inhibitors is GSK1995057; it is a domain antibody (dAb) representing the minimal fraction of antigen-binding units, although it has a low molecular weight (10–13 kDa) is highly stable. One study reported the safety of the administration of GSK1995057 by nebulization; it was evaluated in both non-human primate models of acute lung injury and healthy subjects, suggesting that it could be a therapy for the prevention of acute respiratory distress syndrome [148].

Thus, the previous evidence supports the hypothesis that the TNFR1-selective inhibitors, which do not affect the immunomodulatory functions of TNFR2, have the potential to be used as therapy for diverse inflammatory diseases, including lung affections as COVID-19. Despite the clinical successes of the anti-TNF therapy, reports also indicate that it can induce side effects because TNFR1 mediates cell death; the inflammation process could be a novel drug candidate, although it is necessary to provide pre-clinical data to support further its clinical use.

## 8. Conclusions

Reports have shown diverse effects of anti-TNF agents in the context of COVID-19, some of which are beneficial, whereas others have suggested conflicting results based on the complexity of the nature of TNF and TNFRs pathways. More extensive research may be developed to understand the role of this pivotal axis in the innate immune response against SARS-CoV-2 infection. To support the use of the TNF axis in COVID-19 is imperative to include the dynamics and functions of the TNFRs in the development of selective and efficient therapies. Results suggest that sTNFR1 has an essential role in COVID-19 severity and mortality reason by which must represent a focus of attention in this pathology and a potential therapeutic target in future clinical studies.

## Figures and Tables

**Figure 1 pharmaceuticals-15-00616-f001:**
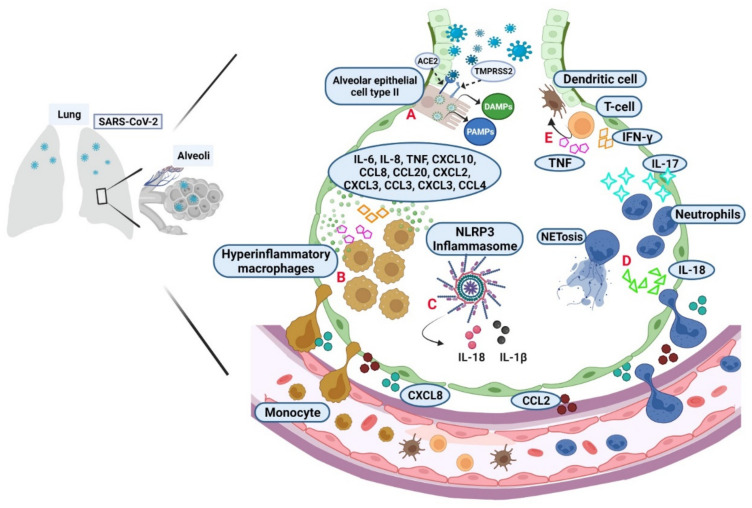
Activation of the immune response during the SARS-CoV-2 infection establishment. (**A**) Alveolar epithelial cell type II express high levels of ACE2 used as the primary cell receptor for SARS-CoV-2 through the S protein activation by the protease TMPRSS2. PAMPs and DAMPs are the first signals to promote innate immune activation. The chemokines CXCL8 and CCL2 promote cell mobilization (CXCL refers to the C-X-C motif chemokine ligand number, where “C” is a cysteine, and “X” represents any amino acid; CCL refers to the C-C motif chemokine ligand number, where “C” is a cysteine). (**B**) In severe COVID-19, there are hyperinflammatory macrophages that promote the release of cytokines and chemokines like IL-6, IL-8, TNF, IFN-γ, IL-1β, CXCL10, CCL8, CCL20, CXCL2, CXCL3, CCL3, CXCL3, and CCL4. (**C**) PAMPs and DAMPs also favor inflammasome activation. The NLRP3 formation is the central platform for caspase-1 activation and the proteolytic maturation of IL-1β and IL-18. (**D**) IL-17 and IL-18 activate neutrophils in the alveolar space evolving into NETosis, which promotes more inflammation. (**E**) Together, all pro-inflammatory signals induce the T cell recruitment, which produces TNF and IFN-γ, and it impacts endothelial and dendritic cells. (Figure created with BioRender.com, adapted from “Cytokine storm template” by BioRender.com. Obtained with https://app.biorender.com/biorender-templates, accessed on 21 April 2022).

**Figure 2 pharmaceuticals-15-00616-f002:**
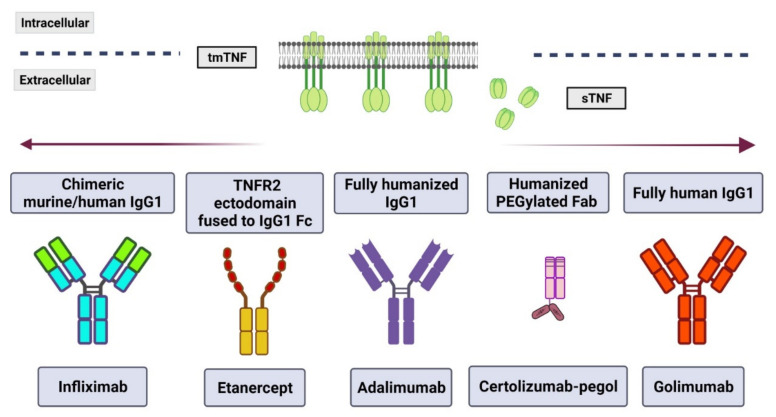
TNF inhibitors approved for therapeutical use. Currently, there are five monoclonal antibodies used in anti-TNF therapy; each one has a specific structure and origin. However, all of them neutralize both transmembrane (tm) and soluble (s) TNF. (Figure created with BioRender.com, accessed on 21 April 2022).

**Table 1 pharmaceuticals-15-00616-t001:** Human TNFR1 inhibitors under progress.

Inhibitor	Class	Description	Properties	Disease Model Evaluation	References
Atrosab	Humanized IgG1	Receptor selective inhibitor	Antagonistic monoclonal antibody	Multiple sclerosis	[134,135]
Fv13.7-Fc	Fab	Receptor selective inhibitor with affinity maturation	Monovalent Fab of Atrosab		[136]
Atrosimab	Fv-Fc1k fusion protein	Receptor selective inhibitor	Fusion protein	Arthritis	[137]
R1antTNF R1antTNF-T8 scR1antTNF	TNF-muteins	Receptor-selective antagonistic activity	Bind to TNFR1 without activation	Acute hepatitis	[138,139,140,141,142]
DMS5541	A bispecific, single variable-domain antibody	Receptor-selective blockade	anti-TNFRI moiety plus an albumin binding moiety	Human Rheumatoid Arthritis	[129]
TROS	Nanobody (Nb) technology	TROS selectively binds and inhibits the TNF/TNFR1 signaling pathway	K_D_ and IC50 values in the nanomolar range	Crohn’s disease ex vivo model; experimental autoimmune encephalomyelitis; multiple sclerosis murine model	[133,143]
PMG (physcion-8-O-β-D-monoglucoside)	A bioactive compound isolated from Chinese herbs	Ligand for TNF receptor from herbal medicines	K_D_ at nanomolar range		[144]
Zafirlukast Triclabendazole (does not compete with a ligand or with PLAD-PLAD assembly *) DS42	Small molecule Small molecule Small-molecule allosteric inhibitor	Small-molecule approaches inhibit receptor interaction or alter receptor conformational dynamics without interrupting ligand binding. Noncompetitive inhibitor without reducing ligand affinity or disrupting receptor dimerization.		Asthma Allergic rhinitis Chronic Idiopathic Urticaria	[145,146]
ASOs	Antisense oligonucleotides	Blocking TNFR1 gene expression		Protection from Radiation-Induced Apoptosis	[147]
GSK1995057	Fully human domain antibody (dAb) fragment	Selectively antagonizes TNF signaling through TNFR1	Phase IIa clinical trial	Respiratory Disorders	[148]
Aptamers AptTNR1		Binding to TNFR1 but not TNFR2	K_D_ around 100 nM		[132,149]

* The impact of the pre-ligand-binding assembly domain (PLAD) has been highlighted in the conformation of active ligand-receptor complexes before TNF association. In fact, soluble TNFRs may interfere with the PLAD-PLAD interaction [117,150].

## Data Availability

Data sharing not applicable.

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
