# Peer review of "Immunosuppressant Therapies in COVID-19: Is the TNF Axis an Alternative?"

_pharmaceuticals, 2022, doi:10.3390/ph15050616_

Round 1

Reviewer 1 Report

This is an interesting review of the relevance of the TNF axis in the treatment of severe COVID-19. Authors appropriately describe the role of TNF and TNF receptors in the immune response to SARS-CoV-2, as well as the available drugs (authorized and in study) that target different cytokines. Also, the authors highlight the potential relevance of targeting TNFR1 in the treatment of severe COVID-19. The manuscript contains comprehensive information and is adequately structured. I only have minor suggestions:

  1. Line 42. Change "curse" to "course".
  2. The sentence in lines 44-46. I think the therapy could improve the clinical outcome of individuals with COVID-19, but this doesn't impact the rapid spread of the virus. Or did the authors refer to vaccines?
  3. Line 78. Please change "infection" to "infectious"
  4. Line 253. I think "patients who lose the ability to respond to therapy" is not the best definition for non-responders patients. The complete sentence is confusing, probably authors can rewrite it. 
  5. Line 279. Change "this" to "these". 
  6. The sentence on lines 330-332 also can be improved. 
  7. I think that adding available information about the status of TNFR1 inhibitors, and preliminary results about the efficacy of safety of the clinical trials including these drugs, could enrich the manuscript and support the conclusions. 

Author Response

Dear Reviewer, thanks for your comments. Below I reply point by point to your suggestions. 

Line 42. Change ...

R= It is done (new line 43).

The sentence in lines 44-46...

R= The sentence has been changed (lines 45-46).

Line 78. Please change...

R= It is done(line 79).

Line 253. I think "patients ...

R= The complete sentence is confusing...

R= The phrase was modified (lines 279-280).

Line 279. Change "this" ...

R= It is done (line 307).

The sentence ...

R= The phrase was modified (lines 370-372).

I think that adding available information about ...

R= In the manuscript's new version, we added section #7 (396-431).

Reviewer 2 Report

The manuscript entitled: Immunosuppressive therapies in COVID-19: is the TNF axis an alternative? is generally interesting but please consider these comments:

1- The title: How many routes lead us to the cytokine storm in the COVID-19 context? needs to be more precise and concise.

2- Line 170: you should explain the basis of activity of the monoclonal antibodies in general first.

3-  Line 181: IL-6R blockade may represent an essential strategy for interrupting the inflammatory process in COVID-19; however, the success has been limited. Why?

4- 185: authorized is duplicated

5- More information should be added regarding the Janus-associated kinase (JAK)-inhibitor, Baricitinib

6- The principle of use of the combination between Baricitinib and remdesivir should be clarified.

7- Line 205: I think the word who is not correct

8- Line 207: the word pleiotropic should be explained

9- I think more detailed information discussing the biological TNF inhibitors should be added.

10- Line 205: I think this title should be divided into titles: one discussing TNF and its inhibitors and the second discussing its role in COVID-19

11- The conclusion section should be more comprehensive.

Author Response

Dear Reviewer, thanks for your comments. Below I reply point by point to your suggestions. 

The title: How many routes lead...

R= In the new version, we were more concise in this phrase (line 67).

Line 170: you should explain ...

R= In the new version, it is included (lines 174-177).

Line 181: IL-6R blockade may represent an ...

R= In the new version is included the probable reasons why it could be (lines 188-195).

185: authorized is duplicated.

R= We apologize for the mistake, and this has been corrected (line 198).

More information should be added regarding...

R= More information was added in this section (lines 215-220).

The principle of use of the combination...

R= More information was added in this section (lines 221-226).

Line 205: I think the word who is not correct.

R= This subhead was modified (line 229).

Line 207: the word pleiotropic should be explained.

R= A phrase was included to explain "pleiotropic" (lines 231-233).

I think more detailed information discussing...

R= More information was added in this section (lines 221-228).

Line 205: I think this title should be divided into...

R= This section was divided into 4 and 5 (lines 229 and 271, respectively).

The conclusion section should be more comprehensive.

R= The conclusion was rephrased (lines 437-441).

Reviewer 3 Report

Palacios et al have shown diverse effects of anti-TNF agents in the context of COVID-19, 354 some of which are beneficial, whereas others have suggested conflicting results based on 355 the complexity of the nature of TNF and TNFRs pathways. I think it is very interesting study. but i recomend the following points

1-more clarification of TNF based therapy side effect

2-more clarification of epigenetic role in TNF therapy

3-English and grammer revision

Author Response

Dear reviewer, thanks for your comments. Below I reply point by point to your comments.

1-more clarification of TNF based...

R= More information was added in this section (lines 322-327).

2-more clarification of epigenetic...

R= Epigenetic information was added in this section (lines 328-336).

3-English and grammer revision

R= We have been more careful about the language.

Reviewer 4 Report

The article is consistent within itself. The references are relevant and recent. The cited sources are referenced correctly. Appropriate and key studies are included. The paper is comprehensive, the flow is logical and the data is presented critically. The scientific level is high, and all the immunological aspects are discussed. Tables and figures are representative.
However, there are some specific comments on the weaknesses of the article and what could be improved:
Specific comments on weaknesses of the article and what could be improved:
Major points - none
Minor points

  1. Consider replacing "inhibitor" with "TNF inhibitor"and additionally - "immunosuppressant"
  2. Line 205 "Who is TNF, and could it be considered a target in COVID-19?" replace "who" with "what"

Author Response

Dear reviewer, thanks for your comments. Below, I reply point by point to your comments.

  1. Consider replacing "inhibitor"...

R= Thanks for your comment, inhibitor was replaced (line 229) and "immunosuppressant" has been inserted in the title.

  1. Line 205 "Who is TNF, and could...

R= Thank you, this subhead was modified (229).